# B-Blockers in Liver Cirrhosis: A Wonder Drug for Every Stage of Portal Hypertension? A Narrative Review

**DOI:** 10.3390/biomedicines12010057

**Published:** 2023-12-25

**Authors:** Dimitrios S. Karagiannakis, Nikolaos D. Karakousis, Theodoros Androutsakos

**Affiliations:** 1Academic Department of Gastroenterology, Laiko General Medicine, Medical School of National and Kapodistrian University of Athens, 11527 Athens, Greece; 2Independent Researcher, 11527 Athens, Greece; karak2727@gmail.com; 3Department of Pathophysiology, School of Medicine, National and Kapodistrian University of Athens, 11528 Athens, Greece; t_androutsakos@yahoo.gr

**Keywords:** non-selective b-blockers, cirrhosis, liver decompensation, spontaneous bacterial peritonitis, hepatorenal syndrome, acute-on-chronic liver failure

## Abstract

In cirrhotic patients, non-selective b-blockers (NSBBs) constitute the reference treatment of choice as monotherapy or combined with band ligation for the prevention of first variceal bleeding and rebleeding, respectively. Furthermore, the last Baveno VII guidelines recommended carvedilol, a b-blocker with additional anti-a1 receptor activity, in all compensated cirrhotics with clinically significant portal hypertension, to prevent liver decompensation. Interestingly enough, NSBBs have been reported to have a potentially positive impact on the short-term mortality of patients with acute-on-chronic liver failure. However, concerns remain about the use of b-blockers in the presence of severe complications, such as refractory ascites, hepatorenal syndrome, spontaneous bacterial peritonitis, or established cirrhotic cardiomyopathy. In addition, it has not been verified yet whether carvedilol supersedes all the other NSBBs in every stage of liver disease, even when severe complications have developed. Therefore, this review aims to illustrate recent data regarding the potential role of b-blockers across all stages of liver disease, beyond the primary and secondary prophylaxis of variceal bleeding, and address the authors’ proposals on the use of NSBBs concerning the severity of liver disease and the patient’s performance status.

## 1. Introduction

Liver cirrhosis is characterized by a blunted synthetic capability in conjunction with an elevated pressure within the portal venous system. Normally, the portal vein pressure ranges between 1 and 4 mmHg higher than the hepatic vein-free pressure and no more than 6 mmHg higher than the right atrial pressure. Higher pressure defines portal hypertension (PH) [1]. The gold standard method for the evaluation of PH is the hepatic vein portal gradient (HVPG), which represents the difference between the wedge hepatic venous pressure and free hepatic venous pressure [2].

At the initial phases of liver cirrhosis, PH occurs due to the obstruction of portal vein intra-hepatic branches by fibrotic tissue and regenerative nodules that progressively replace the healthy liver parenchyma. In addition, potentially reversible intra-hepatic vasoconstriction, induced by the increased production of vasoconstrictors (e.g., endothelins, angiotensin-II, norepinephrine, and thromboxane A2) and reduced release of endothelial vasodilators (e.g., nitric oxide) inside the liver, further aggravates the PH [2]. As a consequence, several portosystemic collaterals, such as the gastro-esophageal varices, are developed to divert blood flow and thereby compensate for the high portal pressure. However, as liver function deteriorates, vasodilating factors are released into the splanchnic vascular network, resulting in systemic splanchnic vasodilation and thus in the worsening of the portal pressure due to an increased portal inflow [3]. Furthermore, splanchnic vasodilation contributes to reduced central blood volume, with central or “effective” hypovolemia. Subsequently, the sympathetic nervous system is activated, the axis of renin-angiotensin-aldosterone is stimulated, and the production of the antidiuretic hormone (ADH) is increased to retain arterial pressure and circulatory homeostasis. However, the activation of these mechanisms predisposes individuals to the development of severe complications, such as variceal bleeding, ascites formation, cirrhotic cardiomyopathy, and hepatorenal syndrome [4,5].

These complications are treated either pharmacologically or invasively regarding their severity. Hence, diuretics and high-volume paracentesis constitute the usual treatment of ascites, whereas non-selective b-blockers (NSBBs) are the gold-standard treatment for the prevention of first variceal bleeding or the prevention of re-bleeding (in combination with band ligation in the latter) [6]. Moreover, NSBBs have been proposed as an effective treatment to prevent liver decompensation. Specifically, the last Baveno VII meeting anticipated the use of carvedilol in patients with advanced compensated cirrhosis and clinically significant portal hypertension (CSPH), defined by HVPG ≥ 10 mmHg, which is the threshold for the development of severe complications [7]. Since then, investigators have characterized b-blockers as “the aspirin” of cirrhosis, in correspondence to the widespread use of aspirin in patients with cardiovascular diseases. 

Nonetheless, many issues need to be clarified before b-blockers are utterly recommended to every cirrhotic patient irrespective of the cirrhosis stage. The aim of this review is to illustrate recent data regarding the potential benefits or negative effects of the use of b-blockers across different stages of liver cirrhosis, taking into account the severity of PH, the existence of complications, and the patient’s performance status. Data on primary and secondary prophylaxis of variceal bleeding will not be mentioned as this topic was recently addressed [8]. 

### 1.1. B-Blockers in Compensated Cirrhotic Patients without or with Small Esophageal Varices (HVPG< or Close to 10 mmHg)

Esophageal varices are present in almost 60% of cirrhotic patients at the time of diagnosis [6]. HVPG ≥10 mmHg is necessary for the development of varices, while the risk of variceal bleeding exists in values of ≥12 mmHg. Variceal bleeding is one of the most serious complications of liver cirrhosis, with an estimated mortality of around 15% [2]. As variceal bleeding mostly depends on variceal size, it is important to avoid variceal enlargement and even better to prevent variceal formation. Although NSBBs are the standard treatment for the prevention of first variceal bleeding, they have failed to impede the development of varices in cirrhotic patients [9]. Interestingly, Groszmann et al. [9]. in a randomized, controlled trial (RCT) of 213 cirrhotics with PH but no varices at baseline showed no significant difference in the development of varices between the timolol and placebo groups after a median follow-up of 4.5 years. Furthermore, the two groups had not significantly different rates of liver-related complications [9]. Regarding the role of NSBBs in inhibiting the enlargement of small varices, an RCT of 206 patients in which 102 were assigned to propranolol and 104 to a placebo showed that 31% of the former and 14% of the latter group had developed large varices after a 2-year follow-up period. However, the limitation of the study was the large number of lost patients during follow-up [10]. In accordance with the previous findings, Sharin et al. [11] confirmed in 2013 the inability of NSBBs to prevent variceal growth [11]. Likewise, a meta-analysis of 6 studies and 784 patients with no or small varices did not demonstrate a significant difference between the NSBBs- and the placebo-treated groups in the degree of small varices’ enlargement. [12]. In contrast, an RCT by Merkel et al. [13] including 161 cirrhotic patients with small esophageal varices found a significantly lower cumulative risk of variceal growth in the nadolol group compared to the placebo group (20% vs. 51%; *p* < 0.001) during a 60-month follow-up period [13]. Additionally, another study favored carvedilol over the placebo in delaying the progression of small varices, as patients in the carvedilol group were found to have an 18% higher probability of not developing large varices while the progression of small to large varices occurred after 20.8 months in the carvedilol group and 18.7 months in the placebo group, respectively [14]. 

Obviously, a discrepancy exists regarding the role of NSBBs in delaying the enlargement of small varices. It is well known that NSBBs reduce the heart rate and the cardiac output by blocking the b1-adrenergic receptors, while they also promote splanchnic vasoconstriction via a b2-adrenergic blockage. These mechanisms result in a reduced portal inflow and, thus, in a decreased portal pressure [15]. However, as previously mentioned, splanchnic vasodilation occurs in the later phases of PH, when the HVPG is ≥10 mmHg and varices have been already developed. Consequently, treatment with NSBBs in this phase could deteriorate the splanchnic vasodilation, reduce the PH, and prevent variceal growth and bleeding. However, in the initial phases of PH, when HVPG is <10 mmHg and varices are absent or HVPG is close to 10 mmHg and varices have likely developed but are still small, treatment with NSBBs would be ineffective as splanchnic vasodilation has not been activated yet and the PH exclusively depends on intra-hepatic portal obstruction and vasoconstriction. Of note, carvedilol, with its additional vasodilating action via the a1 receptors blockage, could potentially play a role in the initial phases of PH by attenuating the intra-hepatic portal vasoconstriction. Further RCTs are necessary to clarify whether the use of carvedilol in the early stages could effectively delay the development or the progression of varices. 

### 1.2. B-Blockers for Preventing Liver Decompensation (Cirrhotic Patients with HVPG > 10 mmHg, but <12 mmHg)

The PREDESCI, a multicenter double-blind RCT, investigated the long-term effect of NSBBs in compensated cirrhotic patients with CSPH without or with small varices. All patients had HVPG measurements and were randomly assigned to be treated with a placebo (101 pts) or b-blockers (100 pts, 67 with propranolol and 33 with carvedilol). The median follow-up period was 37 months, and the primary study end-point was the occurrence of clinical decompensation. The study showed significantly lower decompensating events in the NSBBs arm compared to the placebo arm, (17% vs. 27%, respectively; *p* = 0.041). The most frequent decompensating event was ascites, which presented less frequently in the NSBBs-treated group (9%) compared to the placebo-treated patients (20%) (HR 0.42, 95% CI 0.19–0.92, *p* = 0.03). Concerning secondary end-points, the NSBBs arm had a 40% lower probability of developing large varices (HR 0.60, 95% CI 0.30–1.21) and a 46% lower probability of death (HR 0.54, 95% CI 0.20–1.48) [16]. Based on the above results, the last Baveno VII group recommended the use of carvedilol in compensated cirrhotic patients with CSPH for the prevention of decompensation [7]. It is questionable whether treatment with NSBBs or carvedilol would have good results in compensated cirrhotic patients without CSPH (HVPG < 10 mmHg) as well. Furthermore, it has to be clarified whether non-invasive methods such as liver or spleen elastography could sufficiently substitute HVPG in detecting patients with CSPH, as they have been found incapable of discriminating a significant proportion of patients [7].

### 1.3. B-Blockers in Patients with Decompensated Cirrhosis

Sersté et al. [17] conducted a study in 2010 that raised concerns about the potential deleterious effect of NSBBs in advanced decompensated cirrhotic patients. Specifically, the authors demonstrated that patients with refractory ascites under treatment with propranolol had worse survival compared to untreated patients (5 vs. 20 months, respectively; *p* = 0.0001). Furthermore, the 1-year survival was significantly worse in the propranolol compared to the placebo arm (19% vs. 64%, *p* < 0.0001), while treatment with propranolol was found to be independently associated with poor outcomes in the multivariate analysis. However, as it was observational and not an RCT study, the two groups had no comparative characteristics. Precisely, the propranolol group had more severe underlying liver disease, as defined by a higher Child–Pugh score and a worse biochemical profile. Moreover, the majority of patients in the propranolol group had been treated with unusually high doses of propranolol (≥160 mg per day), which are rarely given in routine clinical practice [17]. The importance of propranolol dose was identified by Bang et al. [18] in a retrospective study with 3719 enrolled patients (3075 with mild and 644 with severe decompensated cirrhosis). The authors reported that patients with mild decompensated cirrhosis treated with propranolol had lower mortality rates compared to the untreated patients (HR: 0.7; 95% CI: 0.6–0.9). Likewise, the mortality was lower in patients treated with propranolol and severely decompensated cirrhosis than in non-treated decompensated patients (HR: 0.6; 95% CI: 0.4–0.9). However, lower mortality was found only when the propranolol dose was lower than 160 mg/day [18]. The potential negative effect of high NSBB doses in advanced decompensated cirrhosis has been attributed to the reduction in blood pressure and cardiac output (via a reduction in heart rate), leading to circulatory incompetence. In the presence of a pre-existed cardiac dysfunction, the risk likely increases. Cirrhosis is often associated with a blunted cardiac function, an entity known as “cirrhotic cardiomyopathy” (CCM), which is characterized by diastolic impairment, reduced systolic reserves during stress, and electrocardiographic alterations, all in the absence of other known causes of cardiac dysfunction [19]. CCM has been found to negatively affect patients’ prognosis [20]. Therefore, an evaluation of cardiac function could be justified before receiving NSBBs, particularly in advanced decompensated states [20]. The interaction between NSBBs and CCM was recently assessed by Giannelli et al. [21] in a retrospective study of 584 pre-transplanted patients. According to that study, refractory ascites (HR 1.52; 95% CI: 1.01–2.28; *p* = 0.0083) and treatment with NSBBs in the co-existence of cardiac dysfunction (HR 1.96; 95% CI: 1.32–2.90; *p* = 0.0009) were associated with an increased waiting list mortality after adjusting for serum sodium and model for end-stage liver disease (MELD) scores [21]. Later on, Koshy et al. [22], in a retrospective study of 319 consecutive patients, investigated whether NSBBs could precipitate the development of major adverse cardiac events (MACE) in a 30-day post-transplantation period. Interestingly, a significantly higher proportion of patients in the NSBBs developed perioperative MACE (32.4% vs. 17.2%, *p* = 0.005). After adjusting for clinical and echocardiographic covariates, NSBBs remained an independent predictor of MACE (OR 2.44; 95% CI: 1.13–5.78), along with pulmonary hypertension, poor functional status, and hepatorenal syndrome [22]. 

In addition to these previous studies, there are many others that have investigated the effect of NSBBs on the prognosis of patients with decompensated cirrhosis, and the majority of them favored the use of NSBBs [23,24,25]. Similarly, in patients with decompensated cirrhosis, Leithead et al. [26] demonstrated a longer median time to death in the NSBBs compared to the non-NSBBs group (150 vs. 54 days, respectively). Moreover, in the multivariate Cox analysis, patients in the NSBBs group had a decreased mortality compared to the propensity-matched non-NSBBs patients (HR 0.55; 95% CI: 0.32–0.95, *p* = 0.032), whereas in cases of refractory ascites, NSBBs were found to be associated with lower mortality (adj HR 0.35; *p* = 0.022) [26]. Subsequently, Ngwa, et al. [27] assessed the role of NSBBs in 90-day post-transplantation prognosis and found a lower 90-day mortality in NSBBs patients (6% vs. 15%; HR 0.27, 95% CI: 0.09–0.88, *p* = 0.03), though these patients had more advanced MELD scores and Child–Pugh scores. However, NSBB use was related to a higher probability of acute kidney injury (AKI) within 90 days (22% vs. 11%, *p* = 0.048). Moreover, 12 of 45 patients (27%) discontinued NSBBs during the follow-up period due to severe hypotension and AKI [27].

Considering a whole number of studies, universal use of NSBBs in advanced decompensated cirrhotic patients, irrespective of cardiac and hemodynamic reserves, cannot be recommended. Nonetheless, in selected cases, with competent cardiac and hemodynamic function, NSBBs in low or moderate doses could potentially improve the outcome.

Regarding carvedilol, a retrospective analysis of consecutive cirrhotic patients with ascites followed for a median time of 2.3 years found better survival between the carvedilol and the non-carvedilol propensity-matching groups (24% vs. 2%; *p* < 0.0001). When stratified according to the severity of ascites, carvedilol provided a 53% lower risk of death when it was administrated in patients with no severe ascites, while in the presence of severe ascites, carvedilol was not found to negatively affect the prognosis [28]. Once again, the dosage of carvedilol seems to be of great importance. Indeed, a current retrospective analysis of 624 patients with ascites demonstrated that the benefit of NSBBs or carvedilol on patients’ survival was diminished when the mean arterial pressure (MAP) was lower than 82 mmHg and disappeared when MAP was < 65 mmHg. Furthermore, in low MAP values, there was a higher tendency towards renal dysfunction [29]. Taking into account the more potent antihypertensive activity of carvedilol, it is becoming obvious that extra caution is needed when high doses of carvedilol are administrated in advanced decompensated individuals. Additionally, there are not many data regarding the role of carvedilol in patients with co-existing CCM. Premkumar et al. [30] recently conducted an RCT, which showed that the combination treatment of carvedilol plus ivabradine reversed left ventricular diastolic dysfunction more often than the placebo. Nonetheless, after 12 months of follow-up, the two groups had similar mortality rates. However, not even one patient in the combination group died when treatment was persistently received. Importantly, non-responders to the combination treatment had worse outcomes (HR 1.3; 95% CI: 1.2–1.8; *p* = 0.046), regardless of age, sex, or MELD score, whereas AKI and hepatic encephalopathy were more frequent in the placebo arm (OR 4.2; 95% CI: 2.8–10.5; *p* = 0.027 and OR 6.6; 95% CI: 1.9–9.7; *p* = 0.04, respectively) [30]. Nevertheless, to date, no studies have investigated whether carvedilol has a greater impact on the survival of decompensated patients with CCM compared to other NSBBs.

### 1.4. B-Blockers in the Setting of Severe Liver-Related Complications (Further Decompensation)

#### 1.4.1. Spontaneous Bacterial Peritonitis (SBP)

In a retrospective study of 607 patients treated with paracentesis for ascites, Mandorfer et al. [31] first identified worse outcomes in patients with SBP taking NSBBs. Specifically, they had significantly more episodes of hepatorenal syndrome (HRS) and AKI and decreased transplant-free survival compared to patients not treated with NSBBs. However, more patients with Child–Pugh stage C and lower arterial blood pressure had been enrolled in the NSBBs group, featuring, once again, the significance of the hemodynamic parameters in advanced decompensated individuals [31]. Bacterial endotoxin has been associated with more severe left diastolic cardiac dysfunction [32] and, consequently, with an increased risk of hemodynamic incompetence. Thus, SBP in patients with low blood pressure and limited hemodynamic consistency predisposes them to the development of HRS and negatively impacts their prognosis. In contrast, in a population of 55 patients with SBP, Lutz et al. [33] found a 30-day post-episode survival of 76% and 41%, respectively, between NSBBs-treated and untreated patients (*p* = 0.049). In addition, during SBP, NSBBs patients had significantly higher fractions of mononuclear cells in ascitic fluid compared to non-NSBB patients (31% vs. 19%, *p* = 0.036), as well as lower IL-8 ascitic concentrations (470 pg/mL vs. 1289 pg/mL, respectively, *p* = 0.29). The authors inferred that NSBBs may attenuate the stimulation of IL-8, leading to more balanced intra-peritoneal inflammation [33]. Similarly, in a subsequent study of 361 patients, NSBBs were found to correlate with improved survival and a lower risk of SBP [18]. The above results are in agreement with those published in 2009 by Senzolo et al. [34] who showed lower SBP rates due to the ability of NSBBs to enhance bowel motility and reduce intestinal permeability [34].

#### 1.4.2. Hepatorenal Syndrome (HRS)

Apart from the aforementioned study of Mandorfer et al. [31], which showed an increased risk of HRS and AKI in advanced cirrhotics treated with NSBBs in the setting of SBP, some other studies reported an increased risk of renal injury in patients taking NSBBs, even in the absence of any infection. Hence, Kalambokis et al. [35] found HRS more frequently in Child–Pugh C subjects treated with NSBBs than in untreated subjects (HRS in 12 months of follow-up: 36% vs. 0%, respectively; *p* = 0.01) [35]. Similarly, in a cohort of 2361 patients waiting for liver transplantation, those who developed AKI (205 pts; median follow-up period: 18.2 months) had ascites more frequently, (79% vs. 51.7%) and more frequent use of NSBBs (45.9% vs. 37.1%; *p* = 0.08) compared to those who had not. Interestingly, NSBBs predisposed them to AKI development when ascites was present (NSBBs plus ascites: HR 3.31; 95% CI: 1.57–6.95), whereas in patients without ascites, the use of NSBBs reduced the risk of AKI (NSBBs without ascites: HR 0.19; 95% CI: 0.06–0.60) [36]. Consequently, the risk from NSBBs is higher in patients with more severe liver disease and more aggravated PH, splanchnic vasodilation, and hyperdynamic circulation. On the other hand, Sasso et al. [37] recently provided results from a retrospective analysis of 529 cirrhotic patients admitted to hospital with AKI between 2015 and 2018. Two hundred and seven patients had been treated with NSBBs and 322 had not. The patients’ characteristics did not differ regarding the MELD score, age, serum sodium, bilirubin, INR, creatinine, blood pressure, history of ascites, or hepatic encephalopathy. However, the former group had a significantly lower platelet count. Patients with NSBBs had less frequent HRS compared to those without (6.3% vs. 12%, *p* < 0.05), but they had pre-renal and cardiorenal AKI more frequently (74.4% vs. 61.5%, respectively *p* < 0.05). Furthermore, patients in NSBBs were less likely to develop SBP or die [37].

#### 1.4.3. Acute on Chronic Liver Failure

Acute on-chronic-liver failure (ACLF) is an excessive inflammatory condition acutely present in patients with an already established liver disease. Acute alcoholic hepatitis, infections, reactivation of chronic hepatitis B, and variceal bleeding are the most common precipitating factors [38]. Patients with ACLF present circulatory incompetence due to extensive vasodilation and poly-organic dysfunction. Regarding NSBBs and their potentially hazardous effect in further deteriorating the hemodynamic status of patients, a sub-analysis of the CANONIC study with 349 hospitalized ACLF patients showed a better 28-day survival in treated- compared to untreated NSBB subjects. In addition, there was a significant difference in the ACLF severity between the two groups. Specifically, the prevalence of ACLF-1 was higher in patients receiving NSBBs, whereas the prevalence of more severe ACLF-2 and ACLF-3 was higher in patients not receiving NSBBs. Moreover, significantly more NSBB patients developed a 1-grade reduction in the ACLF classification (43.2% vs. 27.5%, *p* = 0.0032), while 1-grade worsening was significantly higher in patients not treated with NSBBs (18.1% vs. 10.1%, *p* = 0.0427). [39]. Subsequently, a study of 624 decompensated patients demonstrated that 28-day transplant-free survival was higher in patients treated with NSBBs in general (HR: 0.621; *p* = 0.035), but also in those treated with NSBBs having ACLF in particular (HR: 0.578; *p* = 0.031). Importantly, the survival benefits were markedly attenuated in patients with MAP ≤ 82 mmHg and completely lost in MAP < 65 mmHg. Of note, among patients with MAP ≥ 65 mmHg, NSBBs were consistently associated with improved transplant-free survival, regardless of ACLF presence (HR: 0.480, *p* = 0.034) [29]. Simultaneously, in 2019, another group investigated the role of carvedilol in patients with ACLF. One hundred and thirty-six ACLF patients with HPVG ≥ 12 mmHg were randomized to receive either carvedilol or a placebo. The carvedilol group presented a lower 28-day mortality and lower rates of AKI, SBP, and variceal growth compared to the placebo group (0.6% vs. 24.3%, *p* = 0.044; 3.6% vs. 35.7%, *p* = 0.012; 6.1% vs. 21.4%, *p* = 0.013; 11.1% vs. 32.6%, *p* = 0.021, respectively). Moreover, 2 weeks after the initiation of treatment, a further aggravation of ACLF grading in 22.9% of the controls and 6.1% of the carvedilol-treated patients (*p* = 0.007) was reported, but the benefit was lost at 90 days [40]. The beneficial effect of NSBBs on ACLF has been attributed to their pleiotropic actions, beyond the reduction in portal pressure. In a sub-analysis of the ATTIRE study (Albumin infusion to prevent infection in chronic liver failure), NSBB-treated patients and controls were investigated to find out whether the use of NSBBs was associated with higher hospitalization rates due to infection, more frequent nosocomial infections, higher rates of liver-related complications, or a worse prognosis. Using propensity matching, the two groups had been matched to account for differences in the severity of the disease. According to the results, no differences in renal or cardiovascular dysfunction during days 3–15 of hospitalization were found between the two groups, despite the higher serum creatinine of NSBB patients at baseline. Interestingly, the group of patients under treatment with NSBBs had a significantly lower infection rate and lower serum levels of inflammatory factors, but no significant differences were found between the two groups regarding the rate of nosocomial infections and 6-month mortality. Based on these findings, it could be hypothesized that NSBBs do not prevent nosocomial infections but might downregulate the inflammatory response, preventing ACLF development [41].

## 2. Discussion

Table 1 summarizes the aforementioned studies. An argument exists regarding the potential role of NSBBs across the different stages of liver disease. The retrospective nature of many studies, the heterogenicity between groups, and missing data on HVPG, cardiac function, and hemodynamic parameters are likely the responsible factors. Hence, it would be more reliable to recommend more individualized use of NSSBs based on a patient’s characteristics and performance status, instead of proposing universal guidelines. Concerning the pre-primary variceal prophylaxis, treatment with NSBBs should not be suggested, as according to the RCT of Groszmann et al. [9], NSBBs are ineffective in preventing the development of esophageal varices in patients with HVPG < 10 mmHg [9]. Likely, carvedilol could be more beneficial at these stages, as it inhibits intrahepatic vasoconstriction via its additional anti-a1 receptor activity, but further studies are needed to verify that issue. Regarding the potential role of NSBBs in preventing small varices enlargement, there is conflict among studies. As HVPG measurement, has not been universally performed, it is questionable how comparable the study’s populations are regarding the severity of PH. Nevertheless, it seems that NSBBs could benefit compensated patients with small varices and HVPG over 10 mmHg. Of note, in the randomized carvedilol vs. placebo study of Bhardwaj A et al. [14], where carvedilol significantly delayed the variceal growth, the median HVPG in both groups was >14 mmHg [14].

Carvedilol has also been found to prevent liver decompensation (i.e., ascites formation) in compensated cirrhotics with CSPH, driving the latest Baveno VII to recommend its use in those patients [7,16]. However, it has to be elucidated whether carvedilol is beneficial to all compensated patients with CSPH or exclusively to responders (i.e., those with a sufficient decline in HVPG after the induction of carvedilol). Furthermore, it has to be validated whether non-invasive methods such as the liver stiffness measurement (LSM) or the spleen stiffness measurement (SSM) could adequately substitute HVPG in detecting CSPH, regardless of cirrhosis etiology. Interestingly, the SSM seems to better express the advanced stages of liver cirrhosis when CSPH is present, while the LSM must be combined with other parameters (i.e., platelet count) to achieve the same results [42,43,44]. However, it has to be investigated whether the response to NSBBs could be efficiently identified by changes in the SSM.

After the development of liver decompensation, treatment with NSBBs should not be discouraged, but an individualized approach should be recommended. Unfortunately, studies do not agree with each other about the role of NSBBs in decompensated patients’ mortality. Notably, the majority of the studies are retrospective, with heterogenic populations. Taking into account only RCTs, two studies revealed lower mortality in NSBBs-treated compared to placebo-treated patients [16,40], and one study did not show any significant difference [41]. Considering the sum of the studies published so far, it seems that NSBBs might increase the risk of HRS or AKI and worsen the outcome of patients when given in high doses in cases of severe ascites, Child–Pugh C stage, or CCM. Blood pressure monitoring is of great importance in these patients, and a dose adjustment is crucial, especially when MAP is ≤82 mmHg. In cases of SBP or ACLF, conditions that further deteriorate the hemodynamic stability of patients, NSBBs have been shown to reduce the severity of the inflammatory process and, thus, to prevent the progression to ACLF stage 2 or 3. Moreover, NSBBs have been shown to be capable of potentially improving the short-term mortality of patients with ACLF [29,39,40]. It has been speculated that NSBBs reduce the degree of bacterial translocation by accelerating intestinal peristalsis and reducing intestinal permeability, resulting in a lower production of pro-inflammatory cytokines. Thus, the excessive activation of the inflammatory cascade is avoided [34]. However, in cases of severe ACLF or ACLF progression, when the hemodynamic parameters worsen, NSBBs must be stopped. Figure 1 illustrates our proposal on NSBB use in the different stages of liver cirrhosis.

## 3. Conclusions

In conclusion, when CSPH is absent, NSBBs are likely useless as they cannot prevent variceal formation. The role of carvedilol in this phase needs to be further investigated. When CSPH has been developed, NSBBs, particularly carvedilol, can potentially prevent liver decompensation. In the advanced stages, when liver-related complications are present, the administration of NSBBs should not be discontinued but rather the dose should be adjusted according to the hemodynamic, cardiological, and performance status of patients. In hemodynamically unstable subjects, as defined by low MAP, NSBBs should be given in very low doses or even temporarily stopped. Whether carvedilol is superior to other NSBBs at these advanced stages of cirrhosis needs to be validated. Similarly, it has to be clarified whether carvedilol outweighs the other NSBBs in cirrhotic patients with CCM.

## Figures and Tables

**Figure 1 biomedicines-12-00057-f001:**
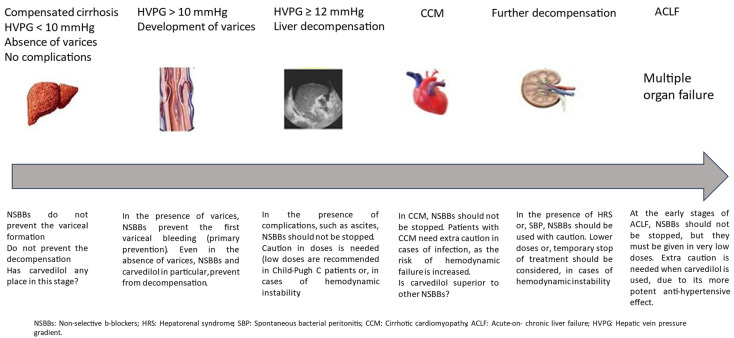
NSBBs across the course of liver cirrhosis.

**Table 1 biomedicines-12-00057-t001:** The effect of NSBBs on cirrhotic patients, beyond the primary and secondary prophylaxis of variceal bleeding.

Authors/Year/Ref	Study	Effect of NSBBs	Results
**Preprimary prophylaxis of variceal formation (HVPG < 10 mmHg)**			
Groszmann RZ, et al. (2005) [9]	RCT	Negative	Timolol vs. placebo in patients without varices Median follow-up 4.5 years No significant difference in variceal formation
**Prevention of small varices enlargement (HVPG > 10 but <12 mmHg)**			
Cales P, et al. (1999) [10]	RCT	Negative	Propranolol vs. placebo in patients with small varices 2-year follow-up period Propranolol did not prevent variceal enlargement
Sharin SK, et al. (2013) [11]	RCT	Negative	Propranolol vs. placebo in patients with small varices 2-year risk of variceal growth: 11% in the propranolol vs. 16% in the placebo group (*p* = 0.786). Variceal bleeding and mortality were comparable between the two groups.
Qi X-S, et al. (2015) [12]	Meta-analysis	Negative	6 studies and 784 patients with no or, small varices. No benefit from the NSBBs, regarding the deceleration of variceal enlargement
Merkel C, et al. (2004) [13]	RCT	Positive	Nadolol vs. placebo 60-month follow-up period Lower risk of variceal growth in the nadolol compared to the placebo group, (20% vs. 51%; *p* < 0.001)
Bhardwaj A, et al. (2017) [14]	RCT	Positive	Carvedilol vs. placebo The carvedilol group had an 18% higher probability of not developing large varices. The mean time of non-progression to large varices: 20.8 months in the carvedilol vs. 18.7 months in the placebo.
**Prevention from liver decompensation; Effect on survival of decompensated patients**			
Villanueva C, et al. (2019) [16]	RCT	Positive	NSBBs vs. placebo Significantly lower decompensating events in NSBBs, compared to the placebo (17% vs. 27%; HR 0.51, 95% CI 0.26–0.97, *p* = 0.041). Ascites development: 9% in NSBBs vs. 20% in the placebo (HR 0.42, 95% CI 0.19–0.92, *p* = 0.03). 46% lower probability of death in NSBBs (HR 0.54, 95% CI 0.20–1.48).
Sersté T, et al. (2010) [17]	Observational Not RCT	Negative	Median survival time: 20 months in non-treated vs. 5 months in propranolol-treated pts (*p* = 0.0001). 1-year probability of survival: 19% in NSBBs treated vs. 64% in untreated patients (*p* < 0.0001).
Bang UC, et al. (2016) [18]	Retrospective	Positive (Only in low doses)	Lower mortality rates in patients with mild decompensated cirrhosis treated with propranolol (HR:0.7; 95% CI: 0.6–0.9). Lower mortality rates in patients with severe decompensated cirrhosis, treated with propranolol (HR:0.6; 95% CI: 0.4–0.9). Reduced mortality, only in doses < 160 mg/day
Gianelli V, et al. (2020) [21]	Retrospective	Negative	Increased waiting list mortality in NSBBs treated, in the co-existence of cardiac dysfunction (HR 1.96; 95% CI: 1.32–2.90; *p* = 0.0009)
Koshy AN, et al. (2020) [22]	Retrospective	Negative	MACE in the 30-day post-transplantation period: 32.4% in NSBBs vs. 17.2% in controls, *p* = 0.005. NSBBs were independently associated with MACE (OR 2.44; 95% CI: 1.13–5.78)
Leithead JA, et al. (2015) [26]	Retrospective	Positive	Patients with ascites awaiting liver transplantation Median time to death: 150 days in NSBBs vs. 54 days in controls. Reduced mortality in NSBBs patients vs. propensity-matched non-NSBBs patients (HR 0.55; 95% CI: 0.32–0.95, *p* = 0.032), In refractory ascites: NSBBs were independently associated with fewer waitlist deaths (adj HR 0.35; 95% CI: 0.14–0.86, *p* = 0.022)
Ngwa T, et al. (2020) [27]	Observational Not RCT	Positive for mortality Negative for AKI	90-day post-transplantation mortality: 6% in NSBBs vs. 15% in non-NSBBs patients; HR 0.27, 95% CI: 0.09–0.88, *p* = 0.03). More AKI in NSBBs (22% vs. 11%, *p* = 0.048).
Sinha R, et al. (2017) [28]	Retrospective	Positive only for patients with mild ascites	Patients with ascites followed for a median time of 2.3 years, Survival: 24% in the carvedilol vs. 2% in the non-carvedilol group (log-rank *p* < 0.0001). A 53% lower risk of death in patients with mild ascites treated with carvedilol. No differences in moderate or, severe ascites
Tergast TL, et al. (2019) [29]	Retrospective	Positive only for hemodynamically competent patients	NSBBs or carvedilol vs. non-treated patients Carvedilol and NSBBs increased the survival, but only in cases of MAP > 82 mmHg
**Effect on patients’ survival in cases of further decompensation (development of SBP, AKI or, ACLF)**			
Mandorfer M, et al. (2014) [31]	Retrospective	Negative	Patients in NSBBs had a worse outcome than the non-treated patients. NSBBs were associated with HRS, AKI, and decreased transplant-free survival.
Lutz P, et al. (2015) [33]	Retrospective	Positive	In 55 patients with SBP 30-day post-episode survival: 76% in NSBBs and 41% in non-NSBBs patients (*p* = 0.049)
Kalambokis GN, et al. (2016) [35]	Retrospective	Negative	HRS more frequently developed in the Child-Pugh C, treated with NSBBs patients than in the untreated patients (In 12 months: 36% vs. 0%; *p* = 0.01)
Kim SG, et al. (2017) [36]	Nested case-control study	Negative	AKI: More frequently in ascites (79% vs. 51.7%) and NSBBs use (45.9% vs. 37.1%; *p* = 0.08) AKI was dependent on the presence of ascites (NSBBs plus ascites: HR 3.31; 95% CI: 1.57–6.95) In patients without ascites, the NSBBs reduced the AKI risk (NSBBs without ascites: HR 0.19; 95% CI: 0.06–0.60)
Sasso R, et al. (2021) [37]	Retrospective	Positive for HRS Negative for cardiorenal AKI	Patients with NSBBs, had less frequent HRS, compared to those without (6.3% vs. 12%, *p* < 0.05), but they had pre-renal and cardiorenal AKI more frequently (74.4% vs. 61.5%, respectively *p* < 0.05).
Mookerjee RP, et al. (2016) [39]	Observational	Positive	Sub-analysis of the CANONIC study, with 349 hospitalized ACLF patients Abetter 28-day survival in treated, compared to untreated with NSBBs patients
Tergast TL, et al. (2019) [29]	Retrospective	Positive only for hemodynamically competent patients	624 consecutive patients with decompensated cirrhosis and ascites. The NSBBs improved the survival in patients with ACLF (HR: 0.578; *p* = 0.031), only when MAP was >82 mmHg
Kumar M, et al. (2019) [40]	RCT	Positive	Carvedilol presented lower 28-day mortality and lower rates of AKI and SBP vs. the placebo. After 2 weeks of treatment: An aggravation of ACLF grading in 22.9% of the controls vs. 6.1% of the carvedilol patients (*p* = 0.007)
Tittanegro T, et al. (2023) [41]	RCT	Neither positive nor negative	No a beneficial impact on the mortality at 28 days, 3, and 6 months from the use of NSBBs

RCT: Randomized controlled trial; NSBBs; Non-selective b-blockers; HR: Hazard ratio; CI: Confidence interval; OR Odds ratio; MACE: Major adverse cardiac events; AKI: Acute kidney injury; MAP: Mean arterial pressure; HRS: Hepatorenal syndrome; SBP: Spontaneous bacterial peritonitis; ACLF: Acute on chronic liver failure.

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
