# Peer review of "B-Blockers in Liver Cirrhosis: A Wonder Drug for Every Stage of Portal Hypertension? A Narrative Review"

_biomedicines, 2023, doi:10.3390/biomedicines12010057_

Round 1

Reviewer 1 Report

Comments and Suggestions for Authors

1- Are there any ongoing or recent clinical trials investigating the role of beta-blockers in cirrhotic patients that the authors have considered in their review? 

2- Could the authors provide a preview of the use of NSBBs concerning the severity of liver disease and patients' performance status proposals and discuss how they align with or deviate from existing guidelines or consensus statements?

3- The authors can use the following reference in the introduction of this manuscript to improve that part. https://doi.org/10.1016/j.ijbiomac.2022.01.134

4- In this review, which specific stages of liver disease are considered, and how the use of beta-blockers may vary or be tailored based on the severity of liver disease?

5-The statement that NSBBs may have a potential positive impact on short-term mortality in patients with acute on chronic liver failure is interesting. Can the authors provide a more detailed analysis of the available evidence supporting this claim and discuss the underlying mechanisms that contribute to this potential positive impact?

Comments on the Quality of English Language

Moderate editing of English language required.

Author Response

Reviewer 1:

  • “Are there any ongoing or recent clinical trials investigating the role of beta-blockers in cirrhotic patients that the authors have considered in their review?”

REPLY: We do not know if other groups of investigators are running any study investigating the role of NSBBs in cirrhotic patients. However, our group has two studies underway. The first one is trying to investigate whether the reduction of HVPG after the induction of NSBBs, can be reflected in equivalent reduction in spleen stiffness measurements. The second one investigates whether carvedilol is superior to other NSBBs in cirrhotic patients with cirrhotic cardiomyopathy. As the two studies are in a preliminary stage, we cannot present any data yet. Nevertheless, in the discussion of this review article, we mentioned in a subtle way the above issues.    

  • “Could the authors provide a preview of the use of NSBBs concerning the severity of liver disease and patients' performance status proposals and discuss how they align with or deviate from existing guidelines or consensus statements?”

REPLY: We added a figure, to provide a preview of the use of NSBBs concerning the severity of the liver disease. Furthermore, in the discussion, we analyzed our proposals for the use of NSBBs across the different stages of cirrhosis. However, the aim of this review was not to drive a change in the current guidelines but to present the latest data on the use of NSBBs, in order to highlight the issues that remain unclarified. Large, randomized controlled studies are needed to answer these issues, leading to modifications of the current guidelines.

  • “The authors can use the following reference in the introduction of this manuscript to improve that part. https://doi.org/10.1016/j.ijbiomac.2022.01.134.”

REPLY: We would like to thank the reviewer for the kind suggestion. Indeed, the above study is very interesting. We tried to modify our introduction based on the introduction of this study. However, this was very difficult, as the content of the above study was different from the content of our study (“Recent progress in polymeric non-invasive insulin delivery”).

  • “In this review, which specific stages of liver disease are considered, and how the use of beta-blockers may vary or be tailored based on the severity of liver disease?”

REPLY: The specific stages of liver disease mentioned in our review are:

-Compensated cirrhosis without or, small esophageal varices (HVPG < or, close to 10 mmHg).  

-Compensated cirrhosis with clinically significant portal hypertension (HVPG > 10 mmHg but still < 12 mmHg; varices have been developed, but no other complications are present).

-Cirrhosis with clinically significant portal hypertension and HVPG > 12 mmHg (Complications such as ascites are present).

-Presence of cirrhotic cardiomyopathy.

-Further decompensation, as defined by the presence of further complications, such as SBP or, HRS

-Acute-on-chronic liver failure.

We added a figure to make more clear which stages of cirrhosis we discussed. In this figure, we have cited our suggestions about the use of NSBBs in different stages of liver cirrhosis, based on the results of the recent studies. Furthermore, we changed the titles of the review sections, as well as the titles of the different sections in Table 1, to make clear the different stages of cirrhosis.

  • “The statement that NSBBs may have a potential positive impact on short-term mortality in patients with acute-on-chronic liver failure is interesting. Can the authors provide a more detailed analysis of the available evidence supporting this claim and discuss the underlying mechanisms that contribute to this potential positive impact?”

      REPLY: We have mentioned the three studies that have shown an improvement in short-term mortality of patients with ACLF under treatment with NSBBs (Ref 29,39,40). We discussed better this issue, to explain the possible underlying mechanism that contributes to the potential positive impact of NSBBs (page 11, lines 401-408). 

Reviewer 2 Report

Comments and Suggestions for Authors

Beta-blockers require a 25-40% reduction in MAP to achieve significant portal pressure reduction. Therefore, while it may be useful in the prevention of esophagogastric varices rupture and as an ongoing non-bleeding countermeasures, is it useful at any point in time? The answer is no. A comparison of the articles mentioned by the author in this review shows a similar trend. In addition, β-blockers sometimes  cause a discomfort during treatment and a re-rebound (Re-Rupture) when treatment is discontinued, and above all, when cirrhosis reaches Child-Pugh CLASS C, the systolic arterial pressure becomes <100 mmHg, and the use of β-blockers in advanced stages of cirrhosis is not very suitable. Therefore, it is not suitable for use in patients with advanced stage cirrhosis. It is therefore inappropriate for use in Every Stage of Portal Hypertension and especially in decompensated cirrhosis.

 Other potential agents that may reduce portal pressure include Carvesirol or, not mentioned here, Angiotensin II Receptor Blockade(ARB) and Terlipressin for AKI.

 However, this paper provides a broad overview of the current global trend in portal hypertension.

    The authors should mention ARBs for prevention varices enlargement   and  Terlipressin for AKI and HRS ,if possible.

Author Response

Reviewer 2:

“Beta-blockers require a 25-40% reduction in MAP to achieve significant portal pressure reduction. Therefore, while it may be useful in the prevention of esophagogastric varices rupture and as an ongoing non-bleeding countermeasures, is it useful at any point in time? The answer is no. A comparison of the articles mentioned by the author in this review shows a similar trend. In addition, β-blockers sometimes  cause a discomfort during treatment and a re-rebound (Re-Rupture) when treatment is discontinued, and above all, when cirrhosis reaches Child-Pugh CLASS C, the systolic arterial pressure becomes <100 mmHg, and the use of β-blockers in advanced stages of cirrhosis is not very suitable. Therefore, it is not suitable for use in patients with advanced stage cirrhosis. It is therefore inappropriate for use in Every Stage of Portal Hypertension and especially in decompensated cirrhosis.”

REPLY: We utterly agree with the reviewer’s statement. NSBBs are not useful at any point in time (i.e. before the formation of varices, when HVPG is < 10 mmHg). Moreover, they may have a hazardous effect on patients with advanced disease and hemodynamic instability. Thus, we removed the phrase “NSBBs are safe for every stage of cirrhosis” on page 12, line 434. However, not every cirrhotic patient with Child-Pugh C has systolic arterial pressure < 100 mmHg, and thus a general recommendation for not giving NSBBs to these patients cannot be supported. That’s why the recent guidelines do not state against the use of NSBBs in these patients. Nevertheless, an individualized approach is necessary, as these patients may develop severe complications, increasing the risk of hemodynamic instability. We believe that we clarified that in the discussion. We also added a figure, showing the different stages of cirrhosis, indicating the necessary caution in the use of NSBBs.

Other potential agents that may reduce portal pressure include Carvesirol or, not mentioned here, Angiotensin II Receptor Blockade(ARB) and Terlipressin for AKI.”

REPLY: The aim of this review was not to discuss portal hypertension and the possible medical interventions against that. The scope was to particularly discuss the thesis of b-blockers in different stages of liver cirrhosis. Thus, the addition of other potential agents that may reduce the portal pressure, such as Angiotensin II receptor blockers, or terlipressin would be out of the scope of this review.

However, this paper provides a broad overview of the current global trend in portal hypertension.

REPLY: We agree with the reviewer. However, the objective of this review was not to provide an overview of the current trend in portal hypertension in general but to provide an overview concerning the role of b-blockers in different stages of portal hypertension in particular.

“The authors should mention ARBs for prevention varices enlargement   and  Terlipressin for AKI and HRS ,if possible.”

REPLY: We would like to thank the reviewer for this interesting suggestion. However, as we mentioned above this would be out of the scope of this review. We did not try to analyze the different medical interventions for liver-related complications, such as variceal enlargement, or HRS. We merely provided data about the role of b-blockers in the treatment of liver-related complications. Nevertheless, as the reviewer’s suggestion is very interesting, we promise to write soon a review article about portal hypertension in general, and the potential therapeutic choices for every portal hypertension-related complication.